# Quantum Photovoltaic Cells Driven by Photon Pulses

**DOI:** 10.3390/e22060693

**Published:** 2020-06-20

**Authors:** Sangchul Oh, Jung Jun Park, Hyunchul Nha

**Affiliations:** 1Qatar Environment and Energy Research Institute, Hamad Bin Khalifa University, Qatar Foundation, P.O. Box 5825 Doha, Qatar; 2Korea Institute for Advanced Study, 85 Hoegiro, Dongdaemun-gu, Seoul 02455, Korea; hyok07@gmail.com; 3Department of Physics, Texas A&M University at Qatar, Education City, P.O. Box 23874 Doha, Qatar; hyunchul.nha@qatar.tamu.edu

**Keywords:** open quantum system, photovoltaic cell, quantum heat engines, quantum thermodynamics, master equations

## Abstract

We investigate the quantum thermodynamics of two quantum systems, a two-level system and a four-level quantum photocell, each driven by photon pulses as a quantum heat engine. We set these systems to be in thermal contact only with a cold reservoir while the heat (energy) source, conventionally given from a hot thermal reservoir, is supplied by a sequence of photon pulses. The dynamics of each system is governed by a coherent interaction due to photon pulses in terms of the Jaynes-Cummings Hamiltonian together with the system-bath interaction described by the Lindblad master equation. We calculate the thermodynamic quantities for the two-level system and the quantum photocell including the change in system energy, the power delivered by photon pulses, the power output to an external load, the heat dissipated to a cold bath, and the entropy production. We thereby demonstrate how a quantum photocell in the cold bath can operate as a continuum quantum heat engine with a sequence of photon pulses continuously applied. We specifically introduce the power efficiency of the quantum photocell in terms of the ratio of output power delivered to an external load with current and voltage to the input power delivered by the photon pulse. Our study indicates a possibility that a quantum system driven by external fields can act as an efficient quantum heat engine under non-equilibrium thermodynamics.

## 1. Introduction

Thermodynamics deals with the evolution of systems, usually in contact with reservoirs, describing the dynamics under universal laws independent of microscopic details. Among its four laws, the second law dictates the total entropy of a closed system can never decrease over time and that the closed system spontaneously evolves toward the state with maximum entropy. One of the possible statements about the second law of thermodynamics is to set the upper bound on the efficiency of heat engines. Heat engines convert heat energy, which typically flows from a hot source to a cold sink, to mechanical energy or chemical energy. The efficiency of energy conversion is defined by the ratio of the work output to the amount of heat energy input. The ultimate efficiency of the heat engine is known in equilibrium thermodynamics to be determined only by temperatures of hot and cold heat baths, $T_{h}$ and $T_{c}$, respectively, that is, $\eta =1-T_{c}/T_{h}$, the so-called the Carnot limit.

Photovoltaic cells (or solar cells) and photosynthesis, just like classical heat engines, convert photon energy from the sun into electric energy and chemical energy, respectively. The upper limit of the efficiency of *p-n* junction solar cells with an energy bandgap is known as the Shockley-Queisser limit [1]. The key assumptions in deriving the Shockley-Queisser limit are (i) photons with energies less than the bandgap are not utilized, (ii) a photon with energy greater than the bandgap produces only one electron-hole pair, and (iii) only the radiative recombination of electron-hole pairs is considered. While the non-radiative loss may be minimized by the manufacturing technology, the radiative recombination is the intrinsic energy loss governed by the law of physics. Assuming the sun and the solar cell are described as black-bodies with temperature $T_{s}=6000\;K$ and $T_{c}=300\;K$, respectively, the maximum efficiency is about 30% for a solar cell with a bandgap of 1.137eV [1]. Shockley and Queisser [1] also showed that the maximum efficiency of a single *p-n* junction solar cell would be approximately 44% around 1.137 eV if there is no radiative recombination loss.

Recently, many theoretical studies have suggested that noise-induced quantum coherence [2,3], Fano-induced coherence [4] or delocalized quantum states of interacting dipoles [5,6,7,8] can reduce the radiative recombination loss of a solar cell, thus enhancing the efficiency of solar cells. The same idea is applied to the photosynthetic complex [9,10,11,12]. Most of these studies employ the donor-acceptor quantum photocell model where the donor is in thermal equilibrium with a hot bath, that is, the sun at 5800K and the acceptor is at room temperature. The photocell operating while continuously contacting both heat reservoirs is called a continuous quantum heat engine [13,14]. A typical example of the continuous quantum heat engine is Scovil and Schulz-Dubois’ three-level masers whose efficiency achieves the Carnot efficiency [15]. By solving the master equation for the quantum photocell, it was shown that the noise-induced quantum coherence or the dark state of the donor may enhance the power output. However, it was not clear whether the efficiency of the photocell could be enhanced by the quantum effect. Some works assumed that the mean photon number of the hot thermal bath could be $\bar{n}=60000$, but the mean photon number of the sun at energy 1.8eV as a black body at 5800K is only about $\bar{n}=0.037$ [16,17]. To address this issue, Reference [16] introduced the pumping term and showed that the dark state could enhance the power output but not its efficiency.

In this paper, we explore another form of a quantum heat engine. We consider two quantum systems, a two-level system and a donor-acceptor quantum photocell, and investigate their quantum dynamics under the coherent driving and the system-bath interaction. Each quantum system is in thermal contact only with a cold reservoir but not with a hot reservoir. Instead, they are driven by a sequence of photon pulses that supplies input energy to the systems, which is conventionally done by a hot reservoir. The photon pulses represent the stream of energy source to the system and may thus remove the unrealistic assumption of the high mean photon number of the sun by the previous works. We solve the time-dependent Markov Lindblad master equation and investigate the thermodynamic quantities such as the change in energy, the heat dissipation to the cold bath, the power delivered by the photon pulse and the entropy generation. Specifically, we introduce the power efficiency of the quantum photocell in terms of the ratio of output power delivered to an external load to input power delivered by the photon pulses.

This paper is organized as follows. In Section 2, we review briefly the quantum dynamics and quantum thermodynamics of an open quantum system based on the master equation approach. In Section 3, we examine a two-level system in a cold bath driven by photon pulses and investigate how the energy, heat current, and entropy change. In Section 4 a quantum photovoltaic cell with donor and acceptor driven by repeated photon pulses is considered. We calculate the quantum thermodynamic quantities and the power output by the sequence of the photon pulses together with engine efficiency. Finally, in Section 5 we summarize our results with some discussion.

## 2. Quantum Thermodynamics of Open Quantum Systems

We start with a brief review of quantum thermodynamics of an open quantum system that exchanges energy and entropy with its environment. The equations presented in this section will be applied to those examples in the next two Section 3 and Section 4. As usual, we assume the Born-Markov approximations: a weak interaction between an open quantum system and an environment, and the extremely short correlation time of the environment, that is, no memory effect. The density operator ρ(t) of the quantum system with a slowly varying time-dependent Hamiltonian obeys the Lindblad-Gorini-Kossakowski-Sudarshan (LGKS) master equation [18,19,20,21,22]
(1)$$\frac{d}{dt}\rho \left(t\right)=L\left[\rho \left(t\right)\right]=-\frac{i}{\hbar }\left[H_{S}\left(t\right),\rho \left(t\right)\right]+D\left[\rho \left(t\right)\right]\;,$$
where $H_{S}\left(t\right)=H_{0}+H_{1}\left(t\right)$ is the Hamiltonian of the system. Here $H_{0}$ represents a time-independent unperturbed Hamiltonian and $H_{1}\left(t\right)$ an external time-dependent perturbation. The decoherence and the dissipation of the open quantum system due to an environmental interaction are described by the non-unitary operator
(2)$$D\left[\rho \right]=\sum_{k}\left(2L_{k}\rho L_{k}^{†}-L_{k}^{†}L\rho -\rho L_{k}^{†}L_{k}\right)\;,$$
where $L_{k}$ are the Lindblad operators determined according to the type of interaction.

From the solution ρ(t) of Equation (Equation 1), one can calculate the quantum thermodynamic quantities. The first law of classical thermodynamics states the energy conservation, dE=δQ+δW. The time-dependent internal energy of the system is given by $E\left(t\right)=\mathrm{tr}\left\{\rho \left(t\right)H_{S}\left(t\right)\right\}$. Its derivative with respect to time gives rise to the first law of quantum thermodynamics [23,24,25]
(3)$$\frac{d}{dt}E\left(t\right)=\dot{Q}\left(t\right)+\dot{W}\left(t\right)=J\left(t\right)+P\left(t\right)\;.$$
Here J(t) is the heat current from the environment into the system
(4)$$J\left(t\right)\equiv \dot{Q}\left(t\right)=\mathrm{tr}\left(\frac{d\rho (t)}{dt}H_{S}\left(t\right)\right)\;,$$
and P(t) is the power delivered to system by external forces,
(5)$$P\left(t\right)\equiv \dot{W}\left(t\right)=\mathrm{tr}\left(\rho \left(t\right)\frac{dH_{S}}{dt}\right)\;.$$
Since $H_{0}$ is the time-independent Hamiltonian, the power can be written as $P\left(t\right)=\mathrm{tr}\left(\rho \left(t\right)\frac{dH_{1}\left(t\right)}{dt}\right)$. The change in energy of the system for finite time can be obtained by integrating the heat current and the power as
(6)$$\Delta E\left(t\right)=Q\left(t\right)+W\left(t\right)=\int_{0}^{t}J\left(s\right)\;ds+\int_{0}^{t}P\left(s\right)\;ds\;.$$

The second law of thermodynamics describes the irreversibility of dynamics, where the entropy plays a key role. The von Neumann entropy S(t) of the system in the state ρ(t) is given by
(7)$$S\left(t\right)=-\mathrm{tr}\left\{\rho (t)\log\rho (t)\right\}\;.$$

The thermodynamic entropy S is written as $S=k_{B}S\left(t\right)$ with the Boltzmann constant $k_{B}$. The net change in the entropy $dS_{\mathrm{net}}$(entropy production) of the whole system+reservoir can be written in terms of the entropy change of the system, dS, and the entropy flow due to heat from an environment to a system, $dS_{e}$, as
(8)$$dS_{\mathrm{net}}=dS-dS_{e}\;.$$

The change in the entropy of the system, dS, over time is written as
(9a)$$\begin{array}{cc} \frac{dS}{dt} & =-\mathrm{tr}\left\{\dot{\rho }\left(t\right)\log\rho \left(t\right)\right\} \end{array}$$
(9b)$$\begin{array}{cc} \; & =-\mathrm{tr}\left\{L[\rho (t)]\log\rho (t)\right\}\;, \end{array}$$
where $\mathrm{tr}\left\{\dot{\rho }\left(t\right)\right\}=0$ and the quantum Markov master Equation (Equation 1) are used. The entropy flow $dS_{e}$ per unit time from the environment into the system is written as
(10a)$$\begin{array}{cc} \frac{dS_{e}}{dt} & \equiv J_{S}=\beta \dot{Q}\left(t\right) \end{array}$$
(10b)$$\begin{array}{cc} \; & =\beta \;\mathrm{tr}\left\{\dot{\rho }H_{S}\left(t\right)\right\}=\beta \;\mathrm{tr}\left\{L\left[\rho \left(t\right)\right]\;H_{S}\left(t\right)\right\}\;, \end{array}$$
where $\beta =1/k_{B}T$ and *T* is the temperature of the environment. The net entropy production rate σ(t) of the system is given by
(11)$$\sigma \left(t\right)=\frac{dS_{\mathrm{net}}}{dt}=\dot{S}\left(t\right)-\beta \dot{Q}\left(t\right)\geq 0\;,$$
where σ(t)≥0 comes from the Spohn inequality [23,24,25,26]. Equation (Equation 11) may be written as
(12)$$\sigma \equiv -\frac{d}{dt}S\left(\rho \left(t\right)\|\rho_{\mathrm{ss}}\right)\geq 0\;,$$
where $S\left(\rho \left(t\right)\;∥\;\rho_{\mathrm{ss}}\right)\equiv \mathrm{tr}\left[\rho \left(t\right)\left(\log\rho \left(t\right)-\log\rho_{\mathrm{ss}}\right)\right]$ is the relative entropy of ρ(t) with respect to the stationary state $\rho_{\mathrm{ss}}$, for example, the canonical state of the system, $\rho_{\mathrm{ss}}=\rho_{\beta }=e^{-\beta H_{s}\left(t\right)}/Z$. This is called the second law of nonequilibrium quantum thermodynamics in the weak coupling limit.

As an application of quantum thermodynamics of open quantum systems presented in Section 2, we first consider a two-level quantum system which is in contact with a cold bath at temperature $T_{c}$ and driven by repeated photon pulses, as depicted in Figure 1. The hot thermal bath supplying energy does not have direct contact with the quantum system. Its role is here replaced by a sequence of photon pulses to the two-level system. We examine how the two-level system absorbs and dissipates energy and generates entropy during this process in order to gain insight into the nonequilibrium dynamics due to photon pulses.

## 3. A Two-Level System Driven by Photon Pulses

The unperturbed Hamiltonian of the two-level system with energy levels $E_{0}$ and $E_{1}$ may be written as
(13)$$\begin{array}{c} H_{0}=-\frac{\hbar \omega_{0}}{2}\sigma_{z}, \end{array}$$
where $\omega_{0}=\left(E_{1}-E_{0}\right)/\hbar$ and $\sigma_{z}=\left|0\rangle \langle 0\right|-\left|1\rangle \langle 1\right|$. The interaction between a two-level system and incoming photon pulses is described by the Jaynes-Cummings Hamiltonian [27]
(14)$$H_{1}\left(t\right)=i\hbar \left[\;g^{*}\left(t\right)\sqrt{\gamma }\;\sigma_{-}-g\left(t\right)\sqrt{\gamma }\;\sigma_{+}\;\right]\;,$$
where $\sigma_{+}=\left|1\rangle \langle 0\right|$ and $\sigma_{-}=\left|0\rangle \langle 1\right|$ are the raising and the lowering operators, respectively. The Jaynes-Cummings Hamiltonian is derived based on the dipole and rotating wave approximations [28]. We set $E_{1}-E_{0}=1\;\mathrm{eV}$. Here γ is the Weisskopf-Wigner spontaneous decay rate
(15)$$\begin{array}{c} \gamma =\frac{1}{4\pi \epsilon_{0}}\frac{4\omega_{0}^{3}d_{01}^{2}}{3\hbar c^{3}}\;, \end{array}$$
where $d_{01}$ is the transition dipole moment between the two states |0〉 and |1〉. A typical value of γ for an atom or a quantum dot for visible light emission is in the order of nanoseconds corresponding to μeV, while the energy at visible frequencies is about eV, that is, femtoseconds. As a numerical calculation becomes demanding with a big difference between these time-scales, we use for our study the values of parameters as listed in Table 1. We consider the photon pulses given at peak times $t_{i}$ as $g\left(t\right)=\alpha \sum_{i}\xi \left(t;t_{i}\right)$ with coherent states having average photon number $\langle n\rangle ={\left|\alpha \right|}^{2}$ and a Gaussian pulse shape [29,30,31]
(16)$$\begin{array}{c} \xi \left(t;t_{i}\right)\equiv {\left(\frac{\Omega^{2}}{2\pi }\right)}^{1/4}\;\exp\left[-\frac{\Omega^{2}{\left(t-t_{i}\right)}^{2}}{4}-i\omega_{0}t\right]\;. \end{array}$$
Here 1/Ω is the pulse bandwidth.

Under the Born-Markov approximation, the interaction of a two-level system and the thermal photon bath is recast to the dissipative operator D acting on the density matrix of the system [21,32]
(17)$$\begin{array}{cc} D_{C}\left[\rho \right] & =\frac{\gamma }{2}\left({\bar{n}}_{c}+1\right)\left(2\sigma_{-}\rho \sigma_{+}-\sigma_{+}\sigma_{-}\rho +\rho \sigma_{+}\sigma_{-}\right)\\ \; & +\frac{\gamma }{2}{\bar{n}}_{c}\left(2\sigma_{+}\rho \sigma_{-}-\sigma_{-}\sigma_{+}\rho +\rho \sigma_{-}\sigma_{+}\right)\;. \end{array}$$

Here ${\bar{n}}_{c}$ is the mean photon number of the cold bath at the frequency $\omega_{0}$ in thermal equilibrium of temperature $T_{c}$
(18)$$\begin{array}{c} {\bar{n}}_{c}=\frac{1}{e^{\hbar \omega_{0}/k_{B}T_{c}}-1}\;. \end{array}$$

As noted in References [32,33], at optical frequencies and room temperature, the mean photon number $\bar{n}$ is very small and negligible while it has a finite value at microwave frequencies and the room temperature. For example, with $\hbar \omega_{0}=\;\mathrm{eV}$ and $T_{c}=300\;K$ one obtains $\bar{n}\approx 6.5\times 10^{-31}$. At the optical frequencies, $\hbar \omega_{0}=1.8\;\mathrm{eV}$ and the temperature of the sun as a black body, $T_{s}=5800\;K$, the mean photon number is $\bar{n}\approx 0.0317$ [16,33]. With this in mind, Equation (Equation 17) reduces to
(19)$$\begin{array}{c} D_{C}\left[\rho \right]\approx \frac{\gamma }{2}\left(2\sigma_{-}\rho \sigma_{+}-\sigma_{+}\sigma_{-}\rho +\rho \sigma_{+}\sigma_{-}\right)\;. \end{array}$$

With Equations (Equation 13), (Equation 17), and (Equation 14), the Lindblad equation for the two-level system, in contact with the cold thermal bath and driven by a Gaussian photon pulse, is given by
(20)$$\frac{d}{dt}\rho \left(t\right)=-\frac{i}{\hbar }\left[H_{0}+H_{1}\left(t\right),\rho \left(t\right)\right]+D_{C}\left[\rho \left(t\right)\right]\;.$$

The quantum dynamics and the quantum thermodynamics of the two-level system are investigated by solving Equation (Equation 20) numerically using the Runge-Kutta method [34]. The parameters used in the numerical simulation are shown in Table 1.

Figure 2 and Figure 3 describe the thermodynamic quantities of the system when a sequence of Gaussian photon pulses are applied at a regular interval (g(t): green curves in (a)) with 〈n〉=1 and 〈n〉=10, respectively. Figure 2a shows the time-evolution of the density matrix elements and the sequence of Gaussian photon pulses, which is first applied around the peak time $\omega_{0}t/2\pi =50$ with 〈n〉=1. The initial state of the two level system is assumed to be in a superposed state $|\psi (0)\rangle =\frac{1}{\sqrt{2}}\left(|0\rangle +|1\rangle \right)$. As shown in Figure 2a, the superposed state decays to the ground state, that is, the system becomes in thermal equilibrium with the cold thermal bath before the photon pulse is applied. When the Gaussian photon pulse is first applied around $\omega_{0}t/2\pi =50$, the system gets excited and then becomes decayed into the ground state after the pulse is gone. This process is repeated according to each Gaussian pulse.

With regard to the first law of quantum thermodynamics, Figure 2b plots the rate of energy change, the power, and the heat current. The heat current J(t) from the environment to the system is always negative. This means the excited state of the two-level system releases its energy to the cold bath. In contrast, the power P(t) and $\dot{E}\left(t\right)$ oscillate out of phase while the photon pulse is applied. Figure 2c shows the entropy S(t) of the two-level system together with its energy E(t), the work done W(t) on the system and the heat transfer Q(t). Figure 2d shows the entropy production $\sigma \left(t\right)=\dot{S}\left(t\right)-\beta \dot{Q}$ as a function of time in relation to the second law of thermodynamics. The entropy production σ(t) is always positive confirming the second law.

In Figure 3, we see more oscillatory behaviors in the quantities due to a stronger photon pulse with 〈n〉=10 than those with 〈n〉=1 in Figure 2. Nevertheless, the overall trend is similar to that explained above for Figure 2.

We now examine how the quantum thermodynamic quantities depend on the temporal shapes of a Gaussian pulse sequence. In Figure 4 and Figure 5, we plot the same quantities as those in Figure 2 and Figure 3, but compare two cases, that is, regularly spaced (Figure 4) and irregularly spaced (Figure 5) sequence of Gaussian pulses with the same mean number $\langle n\rangle ={\left|\alpha \right|}^{2}=1$. As described by the curve g(t), the peak times $t_{i}$’s in Equation (Equation 16) are regularly (not regularly) spaced in the left (right) panel. In both cases, the overall trend of the thermodynamic quantities are similar to that explained for Figure 2 while the actual response of the system does depend on the temporal shape of the pulse sequence. Remarkably, we see that the output power P(t) (blue curves in Figure 4b and Figure 5b) and the accumulated work W(t) (blue curves in Figure 4c and Figure 5c) depend on the temporal shape of the incoming pulses even with the same ${\left|\alpha \right|}^{2}=1$, which can have implications for practical photocell operation. In particular, we find that the case of regular sequencing of pulses yields a higher value of work.

## 4. Quantum Photocell Driven by Photon Pulses

For a quantum heat engine, let us now consider a quantum photovoltaics cell driven by photon pulses, as shown in Figure 6. The quantum photocell we consider is a 4-level quantum system composed of a donor and an acceptor. In 1959, Scovil and Du-Bois [15] consider the 3-level system as the simplest quantum heat engine where one part of the 3-level system is in thermal equilibrium with a hot bath and the other part with a cold bath. Many previous studies [2,3,4,5,6,7,8,9,10] took a similar assumption that the donor of the quantum photocell is in contact with the hot bath, that is, the sun, and the acceptor is in thermal contact with the cold bath. In contrast to the previous works, we assume that the quantum photocell is in thermal contact only with the cold bath. In our previous work [16], the pumping term was introduced in the Lindblad master equation to describe the energy flow from the hot bath. Here the input energy is supplied by the sequence of incoming photon pulses.

The cyclic operation of the quantum photocell can be performed with the sequence as follows: (i) The donor absorbs incoming photons and the electron becomes excited with the transition from the ground state |0〉 to the excited state |1〉. (ii) The phonon vibration makes the excited electron at the donor transfer to the acceptor state |2〉. (iii) The acceptor is coupled to an external load and the current flow (electric work) is represented by the transition decay from the state |2〉 to the state |3〉. (iv) The electron in the state |3〉 of the acceptor returns to the ground state |0〉 of the donor by a vibrational or non-radiative decay.

The unperturbed Hamiltonian of the quantum photocell with 4-levels is written as
(21)$$H_{0}=-E_{0}\left|0\rangle \langle 0\right|-E_{1}\left|1\rangle \langle 1\right|-E_{2}\left|2\rangle \langle 2\right|-E_{3}|3\rangle \langle 3|\;.$$

Similar to Equation (Equation 14), the interaction of the donor of the photocell with the incoming photon pulses is again described by the Jaynes-Cummings Hamiltonian
(22)$$H_{1}\left(t\right)=i\hbar \left[g^{*}\left(t\right)\sqrt{\gamma }\;\sigma_{-}-g\left(t\right)\sqrt{\gamma }\;\sigma_{+}\right]\;,$$
where $\sigma_{+}=\left|1\rangle \langle 0\right|$ and $\sigma_{+}=\left|0\rangle \langle 1\right|$. Same as Equation (Equation 17), the interaction of the donor of the quantum photocell with the cold thermal bath is represented by the Lindblad operator,
(23)$$\begin{array}{cc} D_{c}\left[\rho \right]\; & =\frac{\gamma_{01}}{2}\left({\bar{n}}_{c}+1\right)\left(2\sigma_{-}\rho \sigma_{+}-\sigma_{+}\sigma_{-}\rho +\rho \sigma_{+}\sigma_{-}\right)\\ \; & +\frac{\gamma_{01}}{2}{\bar{n}}_{c}\left(2\sigma_{+}\rho \sigma_{-}-\sigma_{-}\sigma_{+}\rho +\rho \sigma_{-}\sigma_{+}\right)\;. \end{array}$$

The electron transfer between the states |1〉 and |2〉 and that between the state |3〉 and |0〉 are described by the Lindblad operator $D_{\mathrm{ph}}\left[\rho \right]=D_{\mathrm{ph}}^{\left(1,2\right)}\left[\rho \right]+D_{\mathrm{ph}}^{\left(3,0\right)}\left[\rho \right]$, where
(24)$$\begin{array}{cc} D_{\mathrm{ph}}^{\left(i,j\right)}\left[\rho \right]\; & =\frac{\gamma_{ij}}{2}\left({\bar{n}}_{\mathrm{ph}}+1\right)\left(2L_{ij}\rho L_{ij}^{†}-L_{ij}^{†}L_{ij}\rho +\rho L_{ij}^{†}L_{ij}\right)\\ \; & +\frac{\gamma_{ij}}{2}{\bar{n}}_{\mathrm{ph}}\left(2L_{ij}^{†}\rho L_{ij}-L_{ij}L_{ij}^{†}\rho +\rho L_{ij}L_{ij}^{†}\right)\;. \end{array}$$

Here $\gamma_{12}$ and $\gamma_{30}$ represents the transition rate between |1〉 and |2〉 and between |3〉 and |0〉, respectively. $L_{ij}=\left|i\rangle \langle j\right|$ and $L_{ij}^{†}=\left|j\rangle \langle i\right|$ are the lowering and raising operators, respectively. ${\bar{n}}_{\mathrm{ph}}$ is the phonon occupation number at $\hbar \omega =E_{1}-E_{2}=E_{3}-E_{0}$ and $T_{c}=300\;K$. The work done by the quantum photocell to the external load is described by the ohmic dissipation
(25)$$D_{\mathrm{ohm}}\left[\rho \right]=\frac{\Gamma }{2}\left(2L_{3}\rho L_{3}^{†}-L_{3}^{†}L_{3}\rho +\rho L_{3}^{†}L_{3}\right)\;,$$
where $L_{3}=\left|3\rangle \langle 2\right|$. Here Γ represent the conductance of the external load and may be changed from zero corresponding to the open circuit and to a big number representing the short-circuit of the quantum photocell. With Equations (Equation 21)–(Equation 25), the LGKS equation for the quantum photocell is written as
(26)$$\begin{array}{c} \frac{d}{dt}\rho \left(t\right)=-\frac{i}{\hbar }\left[H_{0}+H_{1}\left(t\right),\rho \left(t\right)\right]+D_{c}\left[\rho \right]+D_{\mathrm{ph}}^{\left(1,2\right)}\left[\rho \right]+D_{\mathrm{ph}}^{\left(3,0\right)}\left[\rho \right]+D_{\mathrm{ohm}}\left[\rho \right]\;. \end{array}$$

Since the quantum photocell has no direct interaction Hamiltonian between the donor and the acceptor, the Hamiltonian $H_{S}\left(t\right)=H_{0}+H_{1}\left(t\right)$ of the quantum photocell can be written as the sum of the time-dependent donor Hamiltonian $H_{D}$ and the time-independent acceptor Hamiltonian $H_{A}$,
(27)$$\begin{array}{c} H_{S}\left(t\right)=H_{D}\left(t\right)+H_{A}\;, \end{array}$$
where $H_{D}\left(t\right)\equiv -E_{0}\left|0\rangle \langle 0\right|-E_{1}\left|1\rangle \langle 1\right|+H_{1}\left(t\right)$ and $H_{A}\equiv -E_{2}\left|2\rangle \langle 2\right|-E_{3}\left|3\rangle \langle 3\right|$. This partition makes it possible to express some quantum thermodynamic quantities as the sum of the donor and acceptor parts. The energy of the quantum photocell is given by the sum of the energies of the donor and acceptor
(28)$$\begin{array}{c} E\left(t\right)=\mathrm{tr}\left\{\rho \left(t\right)H_{S}\left(t\right)\right\}=E_{D}\left(t\right)+E_{A}\left(t\right)\;, \end{array}$$
where the donor energy $E_{D}\left(t\right)$ and the acceptor energy $E_{A}\left(t\right)$ are given by
(29a)$$\begin{array}{cc} E_{D}\left(t\right) & ={\mathrm{tr}}_{D}\left\{\rho_{D}\left(t\right)H_{D}\left(t\right)\right\}\;, \end{array}$$
(29b)$$\begin{array}{cc} E_{A}\left(t\right) & ={\mathrm{tr}}_{A}\left\{\rho_{A}\left(t\right)H_{A}\right\}, \end{array}$$
respectively. Here $\rho_{A}={\mathrm{tr}}_{D}\left\{\rho \right\}$ and $\rho_{D}={\mathrm{tr}}_{A}\left\{\rho \right\}$ are the density operators of the donor and acceptor, respectively. Since the photon pulse delivers the power only to the donor, the power P(t) is given by the power of the donor
(30)$$\begin{array}{c} P\left(t\right)=\mathrm{tr}\left\{\rho \left(t\right){\dot{H}}_{S}\left(t\right)\right\}={\mathrm{tr}}_{D}\left\{\rho_{D}{\dot{H}}_{1}\left(t\right)\right\}\equiv P_{D}\left(t\right)\;. \end{array}$$

The heat dissipation occurs at the donor and acceptor. Thus the heat current J(t) is written as the sum of the two parts
(31)$$\begin{array}{cc} J(t) & ={\mathrm{tr}}_{D}\left\{{\dot{\rho }}_{D}\left(t\right)H_{D}\left(t\right)\right\}+{\mathrm{tr}}_{A}\left\{{\dot{\rho }}_{A}\left(t\right)H_{A}\right\}\\ \; & =J_{D}\left(t\right)+J_{A}\left(t\right)\;. \end{array}$$

Finally, the entropy of the quantum photocell S(t)=tr{ρlogρ} can be written as the sum of the entropies of the donor and acceptor, $S_{D}\left(t\right)=-{\mathrm{tr}}_{D}\left\{\rho_{D}\log\rho_{D}\right\}$ and $S_{A}\left(t\right)=-{\mathrm{tr}}_{A}\left\{\rho_{A}\log\rho_{A}\right\}$, too. This is because there is no coherent interaction between the donor and the acceptor so the whole density operator of the system has a structure $\rho =\rho_{D}⊕\rho_{A}$.

We calculate the current through the external load as
(32)$$I=e\Gamma \cdot \rho_{22},$$
and the voltage across the external load
(33)$$eV=E_{2}-E_{3}+k_{B}T_{C}\log\left(\frac{\rho_{22}}{\rho_{33}}\right).$$

The latter comes from the relation $\rho_{22}/\rho_{33}=\exp\left(-\left(E_{2}-E_{3}-eV\right)/k_{B}T_{C}\right)$. The electric power delivered to the external load by the photocell is written as $P_{\mathrm{out}}=I\left(t\right)\cdot V\left(t\right)$ which depends on the external conductance Γ. Now that we have the power delivered by the photon pulse, Equation (Equation 30) and the electric power output $P_{\mathrm{out}}\left(t\right)$, we can define the power efficiency of the quantum photocell as

We solve numerically the LGKS Equation (Equation 26) using the Runge-Kutta method for different sequences of photon pulses to obtain the quantum thermodynamic quantities. Figure 7a plots the population of each level of the quantum photocell as a function of dimensionless time when the sequences of the Gaussian pulses are applied one immediately after another almost in a continuum limit. Figure 7b shows the change in energies of the donor and acceptor, ${\dot{E}}_{D}\left(t\right)$ and ${\dot{E}}_{A}\left(t\right)$, the power $P_{D}\left(t\right)$ delivered to the donor by the photon, the power output $P_{\mathrm{out}}$, the heat dissipation at the donor and the acceptor, $J_{D}\left(t\right)$ and $J_{A}\left(t\right)$. Note that $P_{D}\left(t\right)$ becomes equal to $J_{D}\left(t\right)$ in the steady state. The heat dissipation $J_{D}\left(t\right)$ of the donor is mainly associated with the transfer of electrons from the donor to the acceptor. The heat flow from the donor to the environment is relatively small because the decay rate $\gamma_{01}$ to the environment is much smaller than the electron transfer rate $\gamma_{12}$ from the donor to the acceptor. We note that only the fraction of the transferred heat energy is converted to the power output $P_{\mathrm{out}}$. $J_{D}\left(t\right)$ mainly plays the role of populating the acceptor level 2, which generates the acceptor current for power output as indicated by Equation (Equation 32). It would be interesting to see whether the coherent transfer between the donor and the acceptor may give rise to a different result.

Figure 7c shows the total entropy S(t) of the system and the entropies of the donor and the acceptor, $S_{D}\left(t\right)$ and $S_{A}\left(t\right)$. Our numerical calculation confirms that the total entropy is the sum of those of the donor and the acceptor, $S\left(t\right)=S_{D}\left(t\right)+S_{A}\left(t\right)$, as explained before.

Figure 7d depicts the current I(t), the voltage V(t), and the power efficiency η defined by the ratio of the electric power output $P_{\mathrm{out}}\left(t\right)$ and the power delivered by the photon $P_{D}\left(t\right)$. From the figures, we see that these quantities initially show an oscillatory behavior then become saturated in the long time limit. In particular, the asymptotic value of power efficiency is as high as $\eta_{p}∼0.36$, which can also be interpreted as the work efficiency, that is, work output divided by energy input, when $\eta_{p}$ is constant.

There are different types of quantum heat engines like continuous engine, two-stroke engine and four-stroke engine. Many other studies considered the quantum photocell as the continuous heat engine where the donor is in thermal contact with the hot reservoir and the acceptor is in the cold bath. Figure 7 demonstrates the photocell as a continuous heat engine, which is not in contact with a hot bath, but is supplied input energy by photon pulses. In our case, we have the flexibility of engineering the input photon pulses as desired. In Figure 8, we further compare the power efficiency between the two cases. Figure 8a is the case where the photon pulses are applied at a finite time interval (discrete mode operation). On the other hand, Figure 8b is the case where the photon pulses are applied almost continuously. Both have the same energy parameter 〈n〉=1 of incoming Gaussian pulses. In the discrete mode, we see an oscillatory behavior of power efficiency between 0.2 and 0.6. In the continuum mode, the power efficiency does not oscillate but asymptotically approaches the value $\eta_{p}∼0.36$.
(34)$$\eta_{p}=\frac{P_{\mathrm{out}}\left(t\right)}{P_{D}\left(t\right)}\;.$$

Let us compare the two cases in Figure 7 and Figure 8b both dealing with the continuum limit with different energies 〈n〉=10 and 〈n〉=1, respectively. It is interesting to find that the efficiency turns out to be the same $\eta_{p}∼0.36$ regardless of the pulse energy of our consideration. Of course, the transient behaviors are different as the high-energy case shows an oscillatory behavior while the low-energy case does not. Aside from details, we see that our model of heat engine offers a possibility to make an efficient quantum engine with a proper design.

## 5. Summary

In this paper, we studied quantum thermodynamics of two open quantum systems, the two-level system and the quantum photovoltaic model, driven by the Gaussian photon pulses. By solving the master equation with the time-dependent Hamiltonian of the Gaussian photon pulses, we calculated quantum thermodynamic quantities. For the two-level system in the cold bath, we examined the first law of quantum thermodynamics, which relates the energy change of the system, the heat current, and the power. We also illustrated the second law of thermodynamics by confirming that the entropy production is positive.

More importantly, we investigated the quantum photovoltaic cell in the cold bath driven by the sequence of the Gaussian photon pulses. The power efficiency of the quantum photocell was considered as the ratio of the output power delivered to the external load by the photocell to the input power delivered by the photon pulses. We showed that the quantum photocell as a heat engine can operate both in the discrete stroke mode and in the continuous stroke mode by changing the sequence of the photon pulses.

Our model of quantum heat engine based on a driven quantum system in contact with a single bath seems worthwhile to further investigate. In our work, we showed that the efficiency as high as $\eta_{p}=0.36$ can be achieved, which should be further explored in a broad range of system parameters. The maximum efficiency of the quantum photocell may be obtained by applying the optimal control method [35] or by adopting the quasi-static or quantum adiabatic processes [36,37] There are some meaningful directions to consider. One is to study how the dark state or the quantum coherence can further enhance the performance of the photocell. We also note that recently Chan et al. [31] studied the quantum dynamics of excitons by absorption of single photons in photosynthetic light-harvesting complexes. It would be interesting how the photosynthetic light-harvesting complexes behave when the photon pulses are applied. Moreover, while we considered the Gaussian photon pulses in the current work, other photon pulses, for example, hyperbolic secant, rectangular, or symmetric exponential pulses may be tested to come up with an optimal design [30]. An open problem is how to mimic the thermal photon from the hot thermal bath and to incorporate the thermal photons into the simulation. The quantum photocells considered here can be simulated on quantum computers [38,39] or by using transmon qubits as a working substance [40].

## Figures and Tables

**Figure 1 entropy-22-00693-f001:**
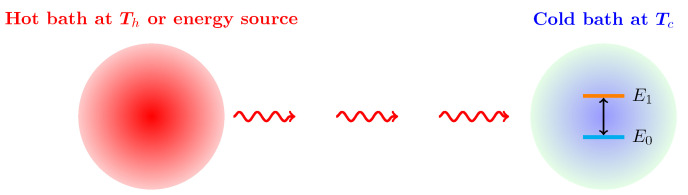
A two-level system with energy levels $E_{0}$ and $E_{1}$ in contact with a cold thermal bath at $T_{c}$ is driven by Gaussian photon pulses serving as an energy source in our work.

**Figure 2 entropy-22-00693-f002:**
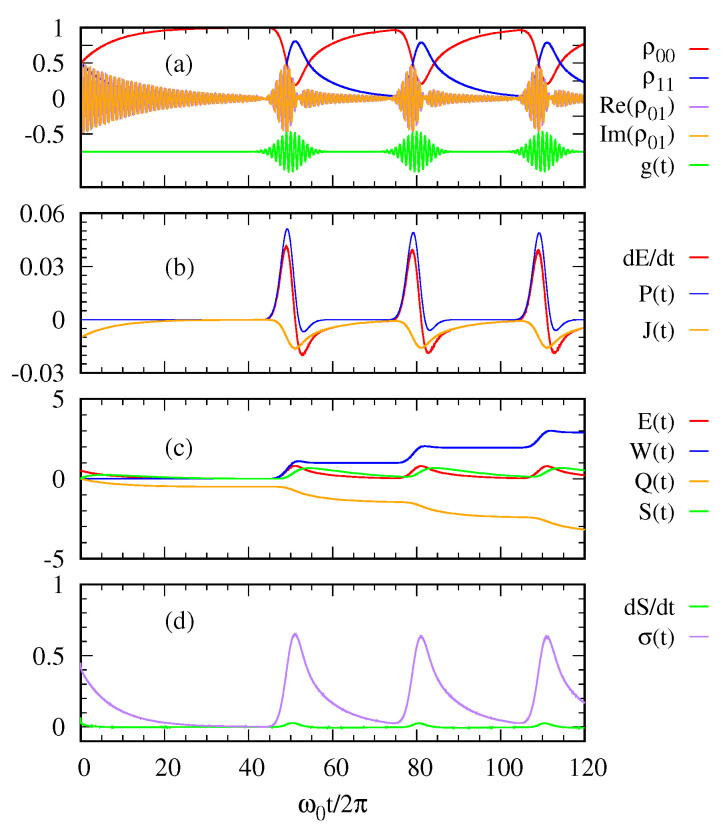
(**a**) The density matrix elements of the two-level system and the sequence of Gaussian photon pulses g(t) are plotted over time. (**b**) The rate of energy change $\frac{dE(t)}{dt}$, the power P(t), and the heat current J(t) are calculated as functions of time. (**c**) The energy E(t), the work W(t), the heat transfer Q(t), and the system entropy S(t) are plotted as functions of time. (**d**) The rate of system entropy change $\frac{dS}{dt}$ and the entropy production σ(t) are plotted over time. The parameters are taken as 〈n〉=1, $\gamma =10^{-2}\omega_{0}$, $\Omega =\omega_{0}/4\pi$, and $\hbar \omega_{0}=1\;\mathrm{eV}$.

**Figure 3 entropy-22-00693-f003:**
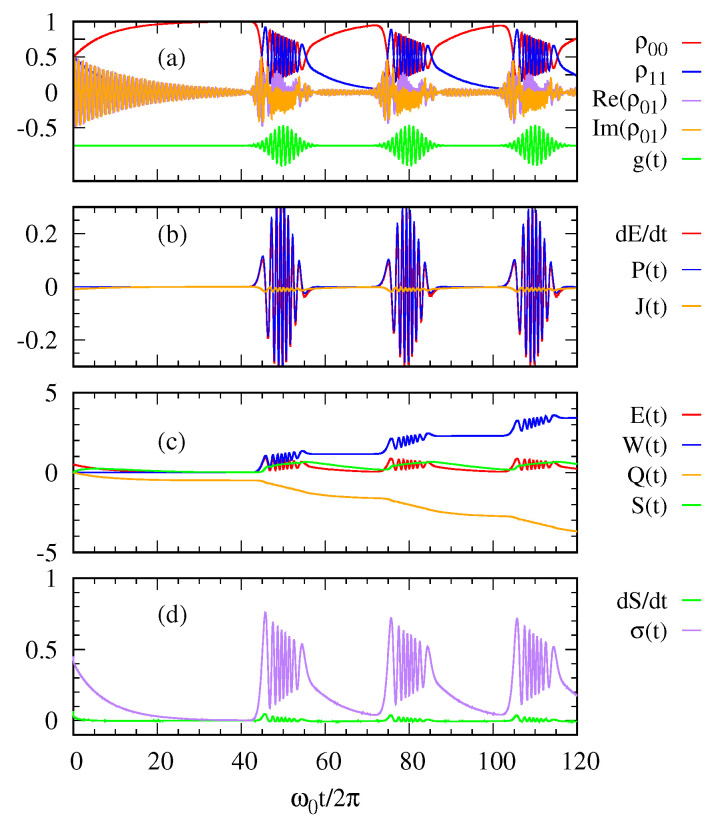
(**a**) The density matrix elements of the two-level system and the sequence of Gaussian photon pulses g(t) are plotted over time. (**b**) The rate of energy change $\frac{dE(t)}{dt}$, the power P(t), and the heat current J(t) are plotted as functions of time. (**c**) The energy E(t), the work W(t), the heat transfer Q(t), and the system entropy S(t) are calculated as functions of time. (**d**) The rate of system entropy change $\frac{dS}{dt}$ and the entropy production σ(t) are shown over time. The parameters are 〈n〉=10, $\gamma =10^{-2}\omega_{0}$, $\Omega =\omega_{0}/4\pi$, and $\hbar \omega_{0}=1\;\mathrm{eV}$.

**Figure 4 entropy-22-00693-f004:**
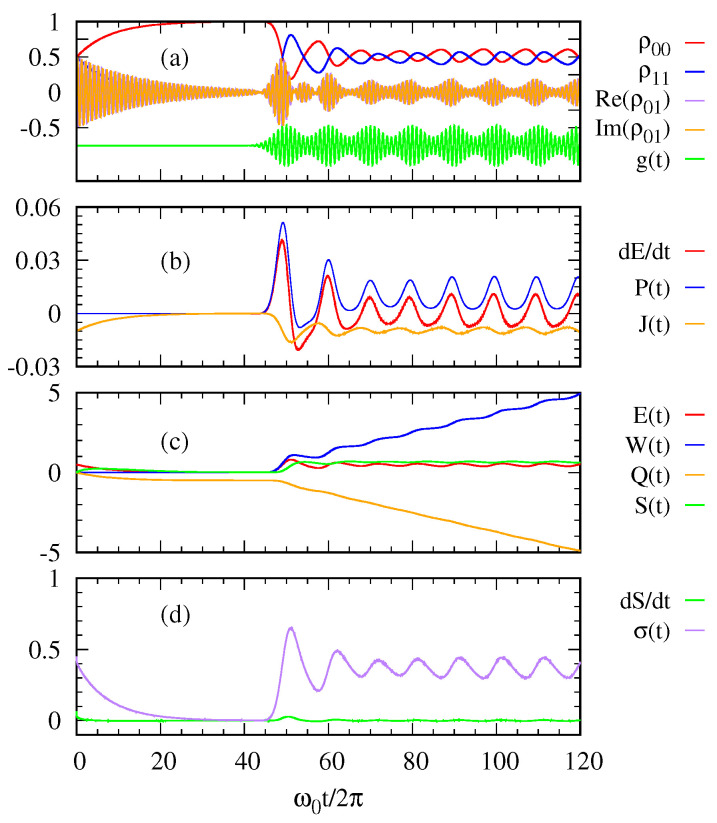
When a Gaussian photon pulse is overlapped with the subsequent Gaussian photon pulse and the interval between them is regular, (**a**) the density matrix elements of the two-level system, (**b**) the rate of energy change $\frac{dE(t)}{dt}$, the power P(t), the heat current J(t), (**c**) energy E(t), work W(t), heat Q(t) system entropy S(t), (**d**) the rate of system entropy change $\frac{dS}{dt}$ and the entropy production σ(t) are plotted as a function of time. The parameters are taken as 〈n〉=1, $\gamma =10^{-2}\omega_{0}$, $\Omega =\omega_{0}/4\pi$, and $\hbar \omega_{0}=1\;\mathrm{eV}$.

**Figure 5 entropy-22-00693-f005:**
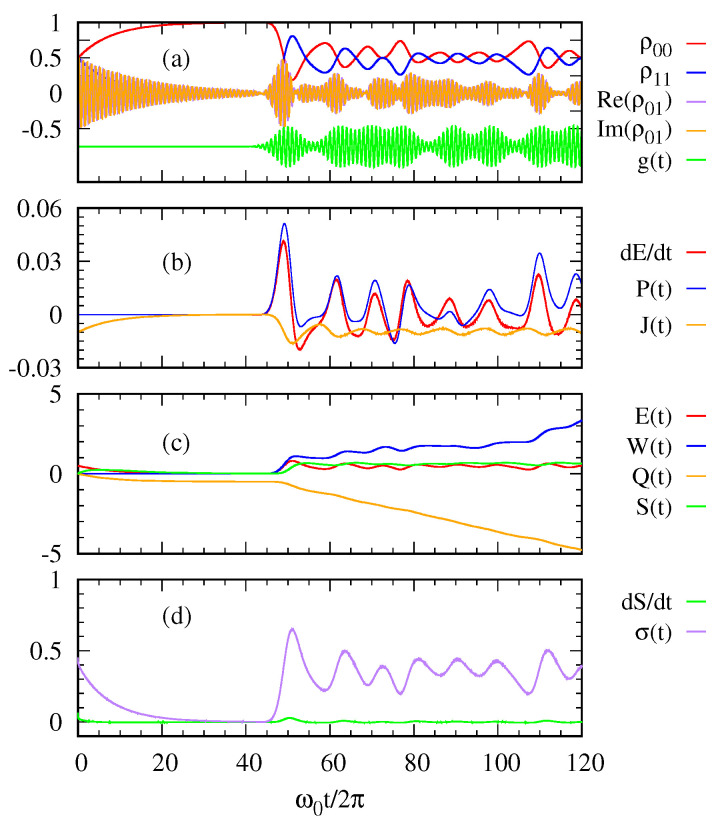
When an irregularly spaced sequence of photon pulses is applied, (**a**) the density matrix elements of the two-level system and the sequence of Gaussian photon pulses g(t), (**b**) the rate of energy change $\frac{dE(t)}{dt}$, the power P(t), and the heat current J(t), (**c**) the energy E(t), work W(t), and heat Q(t), and the system entropy S(t), (**d**) the rate of system entropy change $\frac{dS}{dt}$ and the entropy production σ(t) are plotted as functions of time. Parameters: 〈n〉=1, $\gamma =10^{-2}\omega_{0}$, $\Omega =\omega_{0}/4\pi$, and $\hbar \omega_{0}=1\;\mathrm{eV}$.

**Figure 6 entropy-22-00693-f006:**
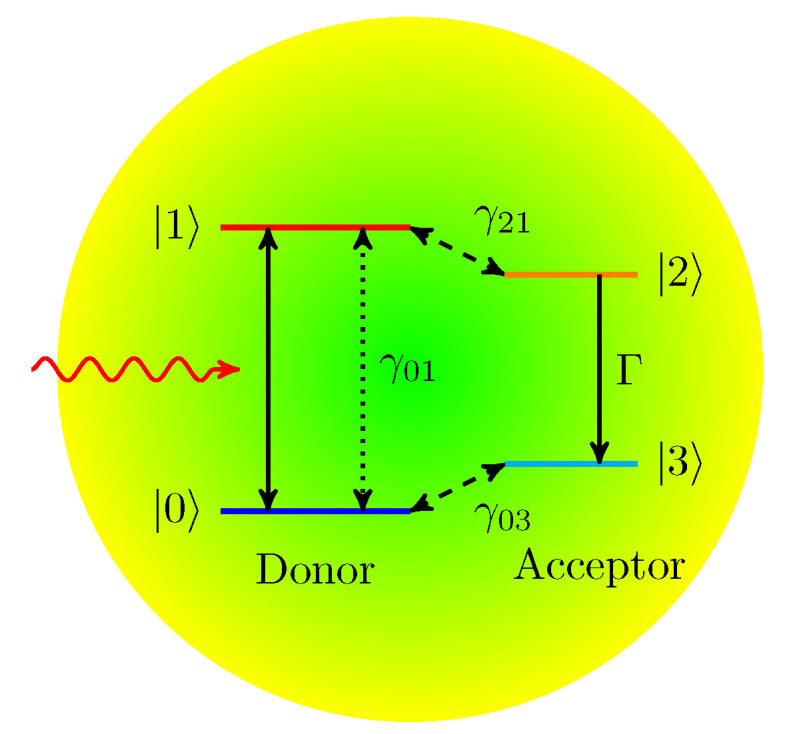
Schematic diagram of a donor-acceptor photocell. $\gamma_{01}$ is the spontaneous decay due to the coupling with the cold thermal bath. $\gamma_{21}$ and $\gamma_{03}$ are the transfer rate between the donor and the acceptor. Γ stands for the external load or electrical resistance.

**Figure 7 entropy-22-00693-f007:**
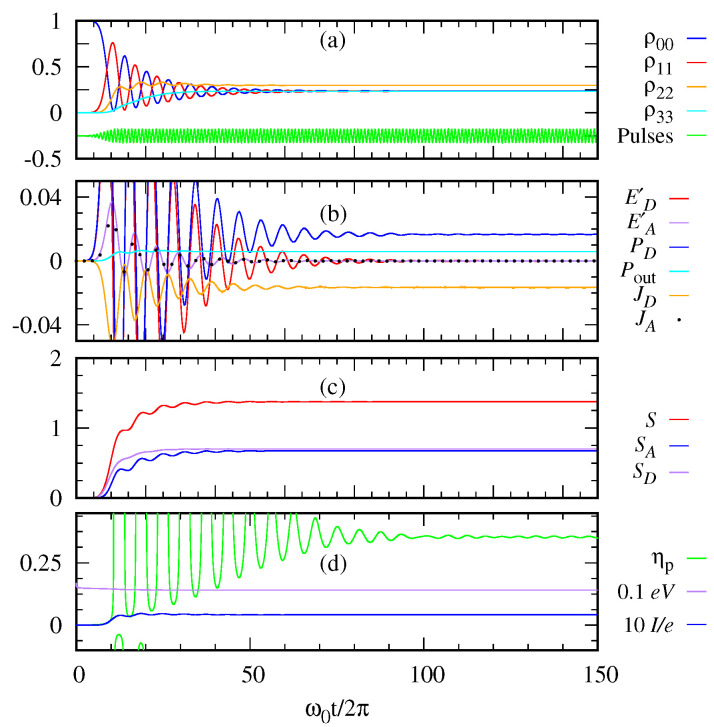
(**a**) The diagonal matrix elements of the density operator of the photocell and the pulse profiles are plotted as a function of time. (**b**) The changes in the energy of the donor and acceptor ${\left(dE/dt\right)}_{D}=E_{D}^{\prime }$ and ${\left(dE/dt\right)}_{A}=E_{A}^{\prime }$, the power delivered to the donor $P_{D}\left(t\right)$ by the photon pulses, the power output $P_{\mathrm{out}}$, and the heat currents of the donor and acceptor $J_{D}$ and $J_{A}$ are plotted as a function of time. (**c**) The entropy of the quantum photocell, S(t), the entropy of the donor, $S_{D}\left(t\right)$, and the entropy of the acceptor, $S_{A}\left(t\right)$, are calculated as a function of time. (**d**) The current I(t), the voltage V(t), and the efficiency η are plotted as a function of time. Parameters: 〈n〉=10, $\gamma_{12}=\gamma_{03}=10^{-2}\omega_{0}$, $\gamma_{01}=10^{-2}\omega_{0}$, $\Gamma =0.1\omega_{0}$, $\Omega =\omega_{0}/4\pi$, and $\hbar \omega_{0}=E_{1}-E_{0}=1.8\;\mathrm{eV}$.

**Figure 8 entropy-22-00693-f008:**
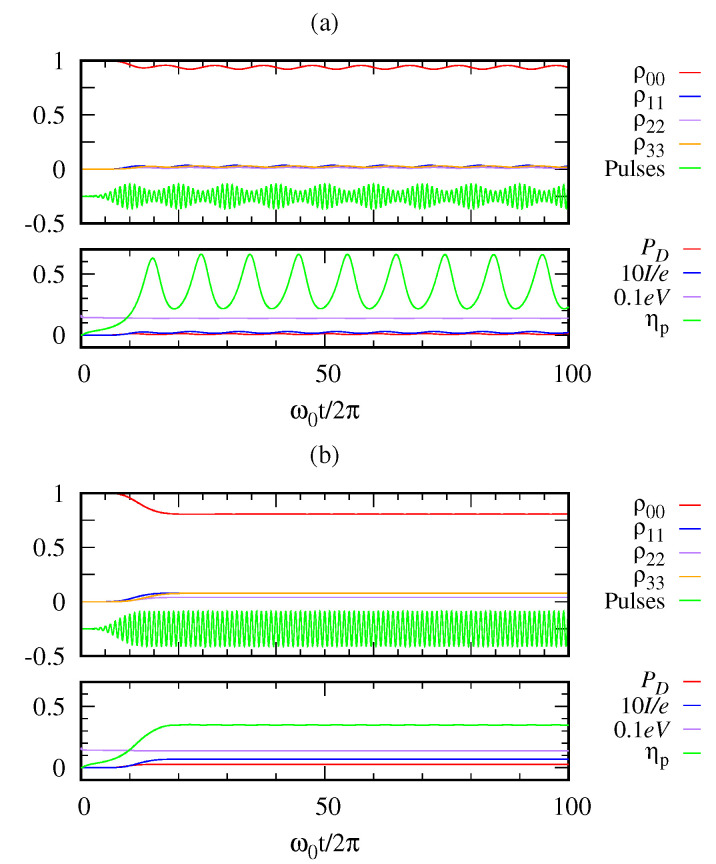
(**a**) From discrete mode to (**b**) the continuous mode operation by changing the interval of the pulses. Parameters: 〈n〉=1, $\gamma_{21}=\gamma_{03}=10^{-3}\omega_{0}$, $\gamma_{01}=10^{-2}\omega_{0}$, and $\Omega =\omega_{0}/4\pi$.

**Table 1 entropy-22-00693-t001:** Typical parameters used in this work.

Energy gap of the two-level system	$E_{1}-E_{0}=\hbar \omega_{0}=1.0$ eV
Energy gap of the donor of the quantum photocell	$E_{1}-E_{0}=\hbar \omega_{0}=1.8$ eV
Energy gap of the acceptor of the quantum photocell	$E_{2}-E_{3}=1.6$ eV
Weisskopf-Winger constant	$\gamma /\omega_{0}=\gamma_{01}/\omega_{0}=10^{-3}∼10^{-6}$
Phonon decay constant	$\gamma_{12}/\omega_{0}=\gamma_{03}/\omega_{0}=10^{-2}∼10^{-3}$
Photon number of a pulse	$\langle n\rangle ={\left|\alpha \right|}^{2}=1\;\mathrm{or}\;10$
Temperature of the cold bath	$T_{c}$ = 300 K
Width of a Gaussian pulse	$\Omega =\omega_{0}/4\pi$

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
