# Peer review of "Quantum Photovoltaic Cells Driven by Photon Pulses"

_entropy, 2020, doi:10.3390/e22060693_

Round 1
Reviewer 1 Report
In this article, the authors study the thermodynamic behavior of two quantum heat engines, a two-level system and a four-level quantum photovoltaic cell. The systems are in contact with a cold bath, while the driving energy is supplied by photon pulses, instead of the traditionally used hot bath. Using numerical simulations, the authors calculate the thermodynamic quantities for pulsed and continuous driving of the engines. For the case of the quantum photocell, they additionally obtain the power efficiency, the ratio of output to supplied power.
The article is in general well-written and the results appear to be quite interesting, especially for the practical implementation of quantum heat engines. For these reasons we believe that the article warrants publication in Entropy, after the authors answer the following observations:
1) In Figs. 4(b), 5(b), it appears that P(t) has the wrong sign. For example, around t=50 (normalized units), it is P(t)<0 and J(t)<0 but dE/dt>0, so the first law (3) is not satisfied. Is seems that in Figs. 4(b), 5(b), P(t) should have the opposite sign than the one shown.
2) In Fig. 7(b), the power delivered to the photocell P_D(t) is not displayed, as erroneously mentioned in the text (it is plotted below in Fig. 7(d)). Back to Fig. 7(b), it seems that only J_D is nonzero (negative, i.e. there is a flow from the system to the environment through the spontaneous emission in the donor). Thus, it appears that all the supplied power P_D is going to J_D, while J_A=0, and note that the acceptor current is associated to the output power. Please clarify this point.
3) In Table 1, the decay rate \gamma is 1.24 micro eV or 124 micro eV, thus close to 1 meV mentioned?
4) In Eq. (33) and the line below, the temperature T is that of the cold bath, T_c?
5) In the caption of Fig. 4, correct the value of \omega_0. In the captions of Figs. 7 and 8, the same symbol \gamma is used twice for all the decay rates.
6) The authors may would like to cite the following relevant works:
(a) A non-thermal quantum engine also using a two-level system
C. Cherubim, F. Brito, and S. Deffner, “Non-Thermal Quantum Engine in Transmon Qubits”, Entropy 2019, 21, 545.
(b) A work using numerical optimal control to find the optimal pulses which maximize the efficiency of the quantum parametric oscillator heat engine. A similar approach might be used to optimize the pulses in the submitted paper
D. Stefanatos, "Optimal efficiency of a noisy quantum heat engine", Phys. Rev. E 90, 012119 (2014).
7) Some typos:
27, “by the temperatures”.
28, “the so-called Carnot limit”
50, “noise-induced”
Captions of Figs. 2 and 3, “the heat current J(t) is calculated as function of”
155, delete double “the”
165, 166, 167, 172, the no. of Fig. 8 is missing.
Reviewer 2 Report
The authors paper titled “Quantum photovoltaic cells driven by photon pulses” studied the quantum thermodynamics of two quantum systems and each driven by a coherent pulse to mimics a photocell working as a quantum heat engine. They analysed the thermodynamics of two open quantum system, the two-level system and the four-level by evaluating the entropy production, power output and work-to-work conversion efficiency of the performance. The topic is interesting for the community of open quantum systems and contribute to understanding of photosynthesisl from physics perspective. However, I find it difficult to understand the clear different between the two systems studied and how it differs from previous studies. Thus, I cannot recommend the paper for publication in its present form.
Here are my concerns ;
For instance, what is the efficiency/performance of the two-level system case?
Why is the two-level case not reaching steady state in a long time?s
In addition, what motivates the choice of Gaussian pulse?
Any specific reason for Runga-Kutta method? It will profit to have a reference as well.
The authors did not discuss the results with respect to other studies in the field. And it will be nice to know if the continuous scheme be map to cyclic heat engine.
In page 12, the reference figures are missing.
Reviewer 3 Report
In this work, the authors present an interesting discussion on the possibility of replacing the hot reservoir with a Gaussian pulse system. The physical system (the working substance) corresponds to a two-level type system immersed in a thermal bath at temperature T_{c} (of an unspecified nature).
The work looks coherent and well developed and explained. It is, therefore, publishable in the journal Entropy.
Respect to section 3:
I want the authors to discuss the inclusion of more energy levels in the model that breaks the approximation presented in equation (14) of the text. Will it be beneficial for the results?
That is to say,
Is it always better to have a working substance with as few levels involved as possible, or is this just an approximation that makes calculations easier (but experimentally feasible)?
In the introduction, it would be advisable to add references of works (when speaking of maximum entropy) that adjusts with closed systems and quantum approximations such as (for example, among others):
- FJ Peña, D Zambrano, O Negrete, G De Chiara, PA Orellana, P Vargas Physical Review E 101 (1), 012116.
- Alvarado Barrios, G .; Peña, F.J .; Albarrán-Arriagada, F .; Vargas, P .; Retamal, J.C. Quantum mechanical engine for the Quantum Rabi model. Entropy 2018, 20, 767.
For example, in lines 160- up to 163, I think that the text needs more references too.
Finally, the authors need to correct Figure 8 when compiles the text.
Round 2
Reviewer 1 Report
The authors have successfully addressed my comments, thus I recommend the publication of the article.
Reviewer 2 Report
The authors have address my questions and concerns in the reversed manuscript. Thus, I recommend the acceptance of the manuscript in Entropy.
Reviewer 3 Report
The authors have answered all the concerns and have produced a second version that can now be published.